# Technical Aspects of Coenzyme Q_10_ Analysis: Validation of a New HPLC-ED Method

**DOI:** 10.3390/antiox11030528

**Published:** 2022-03-10

**Authors:** Abraham J. Paredes-Fuentes, Clara Oliva, Raquel Montero, Patricia Alcaide, George J. G. Ruijter, Judit García-Villoria, Pedro Ruiz-Sala, Rafael Artuch

**Affiliations:** 1Division of Inborn Errors of Metabolism-IBC, Biochemistry and Molecular Genetics Department, Hospital Clínic de Barcelona, 08028 Barcelona, Spain; jugarcia@clinic.cat; 2Clinical Biochemistry Department, Institut de Recerca Sant Joan de Déu, Hospital Sant Joan de Déu, Esplugues de Llobregat, 08950 Barcelona, Spain; clara.olivam@sjd.es (C.O.); raquel.montero@sjd.es (R.M.); rafael.artuch@sjd.es (R.A.); 3Centro de Diagnóstico de Enfermedades Moleculares, Universidad Autónoma Madrid, IdiPAZ, 28049 Madrid, Spain; patricia.alcaide@inv.uam.es (P.A.); prsala@cbm.csic.es (P.R.-S.); 4Biomedical Network Research Centre on Rare Diseases (CIBERER), Instituto de Salud Carlos III, 28029 Madrid, Spain; 5Department of Clinical Genetics, Center for Lysosomal and Metabolic Diseases, Erasmus University Medical Center, 3015 GD Rotterdam, The Netherlands; g.ruijter@erasmusmc.nl; 6August Pi i Sunyer Biomedical Research Institute (IDIBAPS), 08036 Barcelona, Spain

**Keywords:** coenzyme Q_10_, HPLC with electrochemical detection, electrochemical cells, external quality control, ERNDIM

## Abstract

The biochemical measurement of the CoQ status in different tissues can be performed using HPLC with electrochemical detection (ED). Because the production of the electrochemical cells used with the Coulochem series detectors was discontinued, we aimed to standardize a new HPLC-ED method with new equipment. We report all technical aspects, troubleshooting and its performance in different biological samples, including plasma, skeletal muscle homogenates, urine and cultured skin fibroblasts. Analytical variables (intra- and inter-assay precision, linearity, analytical measurement range, limit of quantification, limit of detection and accuracy) were validated in calibrators and plasma samples and displayed adequate results. The comparison of the results of a new ERNDIM external quality control (EQC) scheme for the plasma CoQ determination between HPLC-ED (Lab 1) and LC-MS/MS (Lab 2) methods shows that the results of the latter were slightly higher in most cases, although a good consistency was generally observed. In conclusion, the new method reported here showed a good analytical performance. The global quality of the EQC scheme results among different participants can be improved with the contribution of more laboratories.

## 1. Introduction

Coenzyme Q_10_ (CoQ) is a ubiquitous redox-active lipid in cell membranes and cholesterol transporter lipoproteins. It is composed of a benzoquinone ring and a polyisoprenoid side chain that comprises ten units in humans (Appendix A). CoQ is involved in several biological functions. It plays a critical role in mitochondrial respiratory chain function since it shuttles electrons from complexes I and II to complex III [1]. CoQ also acts as a potent membrane antioxidant and is involved in many other cellular processes, including the modulation of the mitochondrial permeability pores and activation of mitochondrial uncoupling proteins [2].

CoQ deficiencies can be defined by the decrease in CoQ levels in biological samples. Given its essential functions, a deficit in this molecule is associated with various inherited or acquired pathological conditions, which are classified as primary (due to mutations in genes involved in the biosynthesis of CoQ) or secondary (due to mutations in genes unrelated to CoQ biosynthesis or to nongenetic causes) CoQ deficiencies [3,4,5]. Thus, the biochemical demonstration of CoQ deficiency in biological samples has clinical value, since it enables early treatment initiation and may change the natural history of patients with CoQ deficiency [6].

The CoQ content can be analysed in different human biological specimens, where plasma, skeletal muscle, and cultured skin fibroblasts are the most common samples to detect the CoQ deficiency status. The analysis of other samples has also been reported, such as urinary cells, blood mononuclear cells (BMCs), platelets and cerebrospinal fluid (CSF) [7].

The gold standard procedure for biochemical identification of CoQ deficiencies is the measurement of CoQ levels in the above-mentioned biological samples by high-performance liquid chromatography (HPLC). Two main detection systems are commonly used: ultraviolet (UV) and electrochemical (ED) detectors. The first reported methods were based on UV detection, in which CoQ was quantified at 275 nm [8,9]. As ubiquinol (reduced form of CoQ) could not be determined by these procedures because of its low molar absorptivity, HPLC-ED methods were developed [10,11]. Many of them have been reported, using either coulometric or amperometric detectors [12,13]. Some methods based on liquid chromatography-tandem mass spectrometry (LC-MS/MS) have also been reported, with comparable results to HPLC-ED regarding selectivity and sensitivity [14,15]. Other procedures, including fluorescence and chemiluminescence measurements, have also been developed [16,17].

HPLC-ED has significant advantages because it is a more sensitive procedure than UV; thus, it yields more accurate results, and only a minimal sample amount is required. Moreover, it enables the simultaneous detection of both reduced and oxidized forms of CoQ (ubiquinol and ubiquinone, respectively) [18]. For years, many studies relied on the use of the ESA Coulochem ED and their analytical and guard cells for different purposes, including CoQ determination. However, because the production of electrochemical cells was discontinued, this detector is no longer available in most laboratories.

We aimed to standardize a new HPLC-ED procedure for CoQ determination using Thermo Scientific equipment with new electrochemical cells, detail all technical aspects and troubleshooting, and assess its performance in different biological specimens. We also aimed to assess our participation in a new external quality control (EQC) scheme for plasma CoQ determination and compare our results with those obtained using an LC-MS/MS method.

## 2. Materials and Methods

### 2.1. Samples

Samples from patients with suspected and confirmed plasma CoQ deficiencies and controls were used for this study. The following biological specimens were analysed: plasma, skeletal muscle homogenates, urine and cultured skin fibroblasts.

Plasma samples: Ethylenediamine tetra-acetic acid (EDTA) and heparin blood samples were drawn to separate plasma by centrifugation (1500× *g*, 10 min, 4 ºC). The samples were stored at −80 °C until CoQ analysis.

Muscle samples: Skeletal muscle samples were weighed and homogenized with cold SETH buffer (250 mmol/L sucrose, 2 mmol/L EDTA, 10 mmol/L Tris, 5 × 10^4^ IU/L heparin) in an ice bath. Then, the mixture was transferred to a polypropylene tube, followed by vortexing for 2 min, sonication for 5 min, and centrifugation (1500× *g*, 10 min, 4 °C). The supernatant was frozen at −80 °C until CoQ analysis.

Urine samples: A minimum volume of 30 mL of first morning urine samples was collected in standard urine containers, as previously reported [19]. After centrifugation (1500× *g*, 10 min, 4 °C), the urinary pellet was washed with 9 mg/mL saline solution and subsequently centrifuged as before to remove urinary proteins. The urinary pellet was stored frozen at −80 °C until CoQ analysis.

Fibroblasts: Skin fibroblasts were cultured according to standard procedures. The cells were washed twice with 1X PBS and, subsequently, detached from the flasks with trypsin solution. After trypsin inactivation, the cell suspension was centrifuged (500× *g*, 5 min), and the cell pellet was resuspended in 300 µL of PBS. The homogenized cultured fibroblasts were frozen at −80 °C until CoQ analysis.

### 2.2. Method

The CoQ content in different biological specimens was analysed by HPLC using an UltiMate 3000 UHPLC system coupled with electrochemical detection (model ECD-3000RS) (Thermo Fisher Scientific, Waltham, USA). Two coulometric cells were used: 6020RS (preanalytical cell or guard cell) and 6011RS (analytical cell) (Thermo Fisher Scientific, Waltham, MA, USA).

Calibrator (1.16 µmol/L CoQ in ethanol) (Supelco, Darmstadt, Germany), control (reference 0092, Chromsystems, Gräfelfing, Germany), and biological samples were similarly prepared. A flowchart of the general analysis workflow with marks on the timing is depicted in Appendix A. Fifty microlitres of samples were used in all cases, except for urine samples (100 µL). Twenty microlitres of internal standard (5 µmol/L coenzyme Q_9_ (CoQ_9_) in ethanol) (Fluka, Loughborough, UK) were added to all samples. After deproteinization with 500 µL of ethanol, 2 mL of hexane was added to extract both CoQ_9_ and CoQ by vortexing for 10 min. After centrifugation at 1.500× *g* (10 min, 4 °C), the hexane phase was collected, filtered using a 0.22 µm filter, evaporated to dryness under a nitrogen stream, and redissolved in 200 µL of methanol/isopropanol (65:35, *v/v*). The mobile phase consisted of 10 mmol/L ammonium acetate (for LC-MS LiChropur, Merck, Darmstadt, Germany) in methanol/isopropanol (65:35, *v/v*) (Fisher Scientific, Loughborough, UK). Both organic solvents were of electrochemical grade and filtered using a 0.45 µm filter prior to their use. We also tested the mobile phase that our group previously used with the ESA Coulochem II ED, which consisted of 1.06 g/L lithium perchlorate in methanol/ethanol (60:40, *v/v*) [20]. CoQ was separated in a BDS Hypersil C18 column (length and internal diameter: 150 × 3 mm, particle size: 3 µm, Thermo Fisher Scientific); then, it was quantified by ED using CoQ_9_ as the internal standard. The guard cell was set at −700 mV, while the analytical cell was set at −700 mV and +600 mV. The flow rate was 0.5 mL/min, and the injection volume was 30 µL. The total analytical run time was 10 min.

To estimate the cell content of muscle homogenates, urine cell pellet and skin fibroblast samples, the total protein concentration was determined by the Lowry method [21]. CoQ values were normalized to the total protein concentration in those specimens.

### 2.3. Validation Procedure

The following parameters were calculated to validate the method:

(1) The intra-assay precision was assessed using 20 replicates of 2 plasma pools with different CoQ concentrations (0.3 µmol/L and 1 µmol/L). Plasma samples with known CoQ values were used to prepare the pools. Data were reported as the coefficient of variation (CV), which was calculated as follows: (SD/mean)*100, where SD is standard deviation.

(2) The inter-assay precision was calculated by analysing a plasma control material (0.74 µmol/L, reference 0092, Chromsystems) for 10 days. Data were reported as CV.

(3) The linearity, limit of quantification (LOQ) and limit of detection (LOD) were calculated using serial dilutions of a standard solution, which covered the typically observed range of CoQ concentrations (low to high) in cases of CoQ deficiencies and patients under oral CoQ treatment. The LOQ was calculated as (10*SD of the response)/slope, and LOD as (3.3*SD of the response)/slope. The analytical measurement range (AMR) was assessed by analysing ten times five plasma samples covering very low plasma CoQ values (0.12 µmol/L), moderate deficiency (0.3 µmol/L), normal values (0.74 µmol/L and 1 µmol/L), and very high values due to CoQ supplementation (5.6 µmol/L).

(4) The accuracy was assessed in two ways: (a) by comparison of 54 samples with the method previously used by our group with the ESA Coulochem II ED [20], and (b) by laboratory intercomparison for 2 years (2020 and 2021). In 2020, an EQC pilot scheme for plasma CoQ measurement was launched by ERNDIM (European Research Network for Evaluation and Improvement of Screening, Diagnosis and Treatment of Inherited Disorders of Metabolism: www.erndim.org, accessed 9 December 2021). The main goal of this foundation is to provide different quality assurance schemes in laboratory testing for inborn errors of metabolism, and the ultimate objective is to reach a consensus on reliable and standardized procedures for diagnosis [22]. Among different ERNDIM schemes, CoQ was included in the “Special Assays in Serum” (SAS) scheme. Samples were prepared on lyophilized plasma matrices, and four different amounts of CoQ were added in duplicate to cover low, medium and high CoQ values. These amounts were unknown for all participants. In total, 8 samples were sent annually to the participants, who reported their results using the ERNDIM website. Once reported, the results were analysed and different metrological parameters were sent back to the participants at the end of the year, including the accuracy and global interlaboratory variation (see Section 3.2 and Appendix A).

### 2.4. LC-MS/MS Method

The results of the EQC scheme for plasma CoQ determination were compared with those generated from an HPLC-atmospheric pressure chemical ionization-tandem mass spectrometry (HPLC-APCI-MS/MS) method [23]. Briefly, 50 µL of 2.5 µmol/L CoQ_9_ (used as the internal standard) and 50 µL of 2 mg/mL *p*-benzoquinone (used to oxidize all forms of CoQ in the samples) were added to the plasma matrix. After 15 min incubation at room temperature, 850 µL of 1-propanol was added, and the mixture was centrifuged (20,000× *g*, 15 min, 4 °C). The supernatants were transferred to a glass tube and evaporated to dryness under a nitrogen stream. The dried extracts were resuspended in a water/1-propanol (2:8) solution. A calibration curve was prepared with CoQ concentrations ranging from 0.02 to 12 µmol/L. The samples were injected into an Agilent 1290/AB Sciex 4500 LC-MS/MS device. CoQ_9_ and CoQ were separated using a Symmetry C18 HPLC column (Waters, Milford, USA) with a 2-propanol/methanol/formic acid (50:50:0.1) mobile phase and acquired by multiple reaction monitoring (MRM) in the positive mode (CoQ_9_: *m/z* = 796 → 197, CoQ: *m/z* = 864 → 197).

### 2.5. Statistical Analyses

Spearman’s correlation coefficient was calculated using IBM SPSS Statistics for Windows, version 25.0 (IBM Corp., Armonk, NY, USA). Passing–Bablok regression was performed using MedCalc for Windows, version 20.023 (MedCalc Software, Ostend, Belgium).

## 3. Results

### 3.1. Optimization of the Method and Performance in Different Biological Samples

Regarding the chromatographic conditions of the new method, when we first attempted to standardize a procedure for the UltiMate 3000 UHPLC system using a 6011RS electrochemical cell and a mobile phase similar in composition to those previously reported by our group (which contained lithium perchlorate), we observed a dramatic loss of response in the cell after only a few runs. Chromatograms showing this situation are displayed in Figure 1.

Different procedures were performed to recover the cell response, including electrochemical clean-up (application of a potential of +1.000 mV to the cell for up to 10 min with mobile phase flowing) and extensive cleaning of the cell, as indicated by manufacturers, but none of them completely restored the cell performance. Moreover, the use of a preanalytical cell or guard cell (model 6020RS) to electrochemically clean the mobile phase and samples that arrived at the analytical cell, or the exclusive use of electrochemical-grade solvents, did not improve this situation. Thus, we decided to change this mobile phase containing lithium perchlorate to a phase containing ammonium acetate. A gradual loss in cell performance was still observed but was not as remarkable as described before, since it occurred over several runs (Appendix A). The procedure was further optimized after an electrochemical treatment of the cells after each run (Appendix A). This treatment consisted of applying a potential of +1.000 mV to the cells for 10 s and, subsequently, equilibrating them for 10 min (the potentials for each cell are described in Section 2.2).

Once we optimized the new method, we assessed its performance in different biological samples. As an example, typical chromatograms from different specimens (calibrator, plasma, muscle homogenate, urine and skin fibroblasts) are depicted in Figure 2. A head-to-head comparison of 54 samples was used to compare the new procedure with the procedure previously used by our group with the ESA Coulochem II ED [20]. Passing–Bablok regression showed no differences between the two methods: the intercept and slope with their 95% confidence intervals were −0.01(−0.02–0) and 1.03(1–1.06), respectively; the cumulative sum linearity test *p* value was 0.3, which indicates no significant deviation from linearity (Appendix A). Therefore, the data obtained with this new procedure were consistent with the previous procedure, and we maintained our reference values [20,24], which were similar to those reported in the literature by other groups [25,26,27].

### 3.2. Analytical Validation of the Method and EQC Scheme Results

The analytical parameters obtained in the validation study are reported in Table 1.

The accuracy reported in Table 1 was calculated by a comparison of 54 samples with the earlier HPLC-ED method previously used by our group and expressed as % recovery. Concerning the accuracy and laboratory intercomparisons, all ERNDIM SAS scheme reports from 2020 and 2021 were analysed. As an example, a typical report of the results of one sample from 13 participants is shown in Appendix A. Although three methods were used (HPLC-UV-6 laboratories, HPLC-ED-5 laboratories, and LC-MS/MS-2 laboratories) with few participants, a good consistency among the laboratories was observed in that individual report. The global performance of all participants in 2020 and 2021 is shown in Table 2. Details of how different metrological variables were calculated are presented in Appendix A. For example, the accuracy was reported as the mean outcome of the eight samples. In our case, the mean values detected (2020—0.860 µmol/L, 2021—1.34 µmol/L) were consistent with those reported from all participants (Appendix A). Since different methods were used, we can consider that the accuracy of our method (calculated as % deviation) was acceptable. In general, the quality of the other results from all participants in the EQC scheme was not optimal, including a low precision and recovery. However, a remarkable improvement in interlaboratory CV was observed in 2021 in comparison to the 2020 performance.

### 3.3. Method Comparison

The results of the EQC scheme obtained with the new HPLC-ED method (Lab 1), HPLC-APCI-MS/MS procedure (Lab 2), and from all participants are stated in Table 3 and depicted in Figure 3. The results of one sample (SAS2021.06) were not reported by Lab 1, since interferences were observed in the chromatogram.

Due to the low number of samples analysed, Spearman’s correlation coefficient was used to search for correlations between the two methods and between each method and the results reported by all participants. A significant positive correlation was observed in all cases (HPLC-ED vs. LC-MS/MS: *r* = 0.842, *p* < 0.001; HPLC-ED vs. all participants: *r* = 0.979, *p* < 0.001; and LC-MS/MS vs. all participants: *r* = 0.873; *p* < 0.001). In most cases, as depicted in Figure 3, the results obtained with the LC-MS/MS method were slightly higher than those obtained with the HPLC-ED procedure, although a good concordance was generally observed between them.

## 4. Discussion

CoQ levels can be analysed in different human biological samples. Skeletal muscle has been considered the gold standard for investigating CoQ status for many years, but it has limitations, the most significant limitation being that muscle biopsy is an invasive procedure, and it cannot be used for treatment monitoring purposes. Fibroblasts can be obtained from a much-less-invasive procedure, and they enable functional studies to be performed. However, they also have limitations: some false negative results have been reported in patients with muscle CoQ deficiency [28]. Urine is a non-invasive sample, and it has been suggested to be useful as a surrogate biomarker of the kidney CoQ status, although this correlation remains to be demonstrated. Plasma has several advantages, since it represents a minimally invasive sample, and decreased levels of plasma CoQ may reliably indicate secondary deficiencies. However, plasma CoQ can be influenced by both dietary intake and lipoprotein concentrations; thus, it is not recommended for the diagnosis of primary CoQ deficiencies. The measurement of CoQ in lymphocytes can also be useful for diagnosis if the patient has not received CoQ supplementation [29]. Regarding CoQ treatment monitoring, we previously reported that among a wide array of biological samples, plasma appeared to be the best specimen for this purpose, since the CoQ determination in cells (such as BMCs, platelets and urinary cells) requires extensive preanalytical preparation and may display high biological variation [24].

HPLC-ED is both selective and sensitive, which makes it an ideal tool to measure low levels of analytes in complex matrices. Selecting specific operating potentials can result in the selective detection of a compound from a multicomponent mixture. This technique relies on the electroactive nature of compounds that can be oxidized or reduced, and it provides highly sensitive and selective analyses for a wide range of biological and pharmaceutical compounds. Our group reported a procedure for CoQ determination in serum by HPLC-ED almost two decades ago [30]. Since then, we reported CoQ determination in other biological specimens, including muscle, fibroblasts, urine, platelets and BMCs [19,20,24]. All of these procedures were based on the use of the ESA Coulochem II ED. However, since the production of the electrochemical cells used with this detector was discontinued some years ago, a new procedure for CoQ determination using new equipment was necessary.

All previous procedures for CoQ determination that our group and other authors reported used a mobile phase containing lithium perchlorate, which is an electrolyte salt with strong oxidizing properties. Thus, the dramatic loss of response that we observed in the new cells (Figure 1) may result from the chemical damage of the electrodes, as noted by Lee et al. [31]. A previous report used this approach, but few samples were analysed [32]. This application may work under these conditions, as the authors demonstrate, but it is probably not reliable for the clinical analysis of large series of CoQ analyses in different biological samples. The use of ammonium acetate, which is a salt routinely used in many laboratories for different HPLC-ED and mass spectrometry procedures [33,34], instead of lithium perchlorate, was essential for optimizing the procedure of CoQ determination. It is well known that the use of highly pure solvents and chemicals in the mobile phase is crucial for minimizing the contamination of the ED [35]. In our case, filtered electrochemical-grade solvents and ammonium acetate of the highest purity available must be used to optimize the procedure and avoid the loss of response in the cell. Trace levels of electroactive impurities may significantly increase background noise and cause permanent cell damage. Thus, it is of great importance that the solvents and samples are filtered before use and a filter must be placed before the flow-through cells. In our case, two graphite filters were used at pre- and post-injector locations (models 70–0898 and 70–3824, respectively).

The 6011RS coulometric cell contains two working electrodes, which provide the surface area for the electrochemical reaction. In this model, the electrode surfaces are smaller than those of the cells in the ESA Coulochem II ED, so we hypothesize that its lifetime is shorter, i.e., it allows shorter periods of operation without a reduction in signal. In the analytical cell, components are reduced at the first electrode and, subsequently, oxidatively measured at the second electrode. This reversal is employed to improve selectivity since many common interferences are greatly reduced or eliminated. However, the use of a preanalytical cell or guard cell is recommended because it helps eliminate many electroactive, undesirable and interfering compounds from subsequent detection in the analytical cell. In our case, the electrochemical clean-up of the cells after every single run was also essential to maintain the cell performance over time (Appendix A).

An analytical validation of the present method was assessed after calculating different validation parameters in Section 3.2 and demonstrated a good analytical performance. The obtained CVs were lower than 10%, which is appropriate for the clinical assessment of the CoQ status. Indeed, the procedure was accredited by ENAC (Spanish National Accreditation Body) according to the UNE-EN ISO 15,189 norm for medical laboratories. Furthermore, the AMR covered a wide range of CoQ concentrations, including severe deficiencies or high CoQ values due to oral CoQ treatment. Regarding medical decision limits, there is no general consensus. We can consider the lowest limit of the reference interval (around 0.4 µmol/L) the limit to detect plasma CoQ deficient status. This value is inside the AMR for the developed method. Regarding the highest CoQ values, there is not a clear medical decision limit since these values are only observed when patients are under CoQ supplementation. Once again, there is no consensus about which plasma CoQ values are advisable for treatment monitoring. The total analytical run time was 10 min for all types of biological specimens. Although the reported method is faster than our previously reported procedures for CoQ determination [24], we had to electrochemically clean the cells after every run to improve the lifetime of the cells; thus, the final total run time (20 min) was equivalent to our previous methods.

Regarding the ERNDIM EQC scheme results, the global quality was not good. While individual reports from all participants usually offered good results, as observed in Appendix A, the global performance assessed at the end of every year presented some low-quality outcomes, as shown in Table 2. The most plausible explanation for this is that this scheme was new, and few participants were involved. More than 200 laboratories participated in some ERNDIM schemes, but only 11 and 13 participants were registered for plasma CoQ determination in 2020 and 2021, respectively. Furthermore, three different technologies were employed, which probably increased the variability among laboratories. In this regard, we compared our results (Lab 1) with those reported by (Lab 2), as LC-MS/MS methods represent the most widely used assays in recent years for the study of inborn errors of metabolism. Only two participants reported using an LC-MS/MS method in the 2021 scheme, which makes it difficult to draw a conclusion, although we observed slightly higher results for this procedure (Lab 2) than our HPLC-ED method (Lab 1). Other issues deserve further investigation, such as CoQ instability in the lyophilized matrix, which may explain the low overall analyte recovery, or the presence of interferences that can mainly affect UV and ED systems. In any case, one of the most important metrological variables, which is the global interlaboratory variation (calculated as CV), was significantly reduced from 2020 to 2021 (Table 2).

Some limitations of our study include the fact that the validation of the method was carried out for plasma samples and that we did not study other characteristics, such as the matrix effect. Although we assessed the performance of the new method in different biological specimens, including all specimens that are usually analysed in clinical laboratories for diagnosis and monitoring purposes, the validation study was performed only for plasma samples due to the difficulties of having large series of muscle and fibroblasts samples. However, this method has proven to be useful to detect primary CoQ deficiencies in muscle [36], and especially for treatment monitoring purposes, where plasma samples are thought to be the best biological samples [24]. Finally, a low CoQ solubility and stability and the use of adequate matrices in calibrators and controls are critical issues that deserve further investigation.

## 5. Conclusions

We report the validation of a new HPLC-ED method for CoQ analysis in plasma samples that can be used for CoQ analysis in other biological samples using a Thermo Scientific apparatus. The optimization of this method shows that the mobile phase containing lithium perchlorate was not appropriate for the analysis of a large series of samples, since the lifetime of the electrochemical cells was brief, at least in our study. The analytical performance of the present method was successful. We also compared the results of a new EQC scheme for the plasma CoQ determination using our method and an LC-MS/MS procedure. Regarding the EQC scheme performance among different participants, some degree of improvement was observed in 2021, but the participation of more laboratories appears to be mandatory to ascertain the better quality and utility of this new ERNDIM scheme.

## Figures and Tables

**Figure 1 antioxidants-11-00528-f001:**
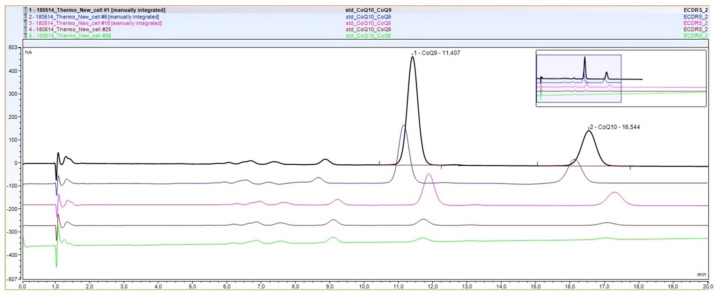
Chromatograms of a calibrator solution (which contained coenzyme Q_9_ (CoQ_9_) and coenzyme Q_10_ (CoQ)) showing the loss of response in the 6011RS electrochemical cell. The same sample was injected 6 (blue), 15 (pink), 25 (brown) and 35 (green) times.

**Figure 2 antioxidants-11-00528-f002:**
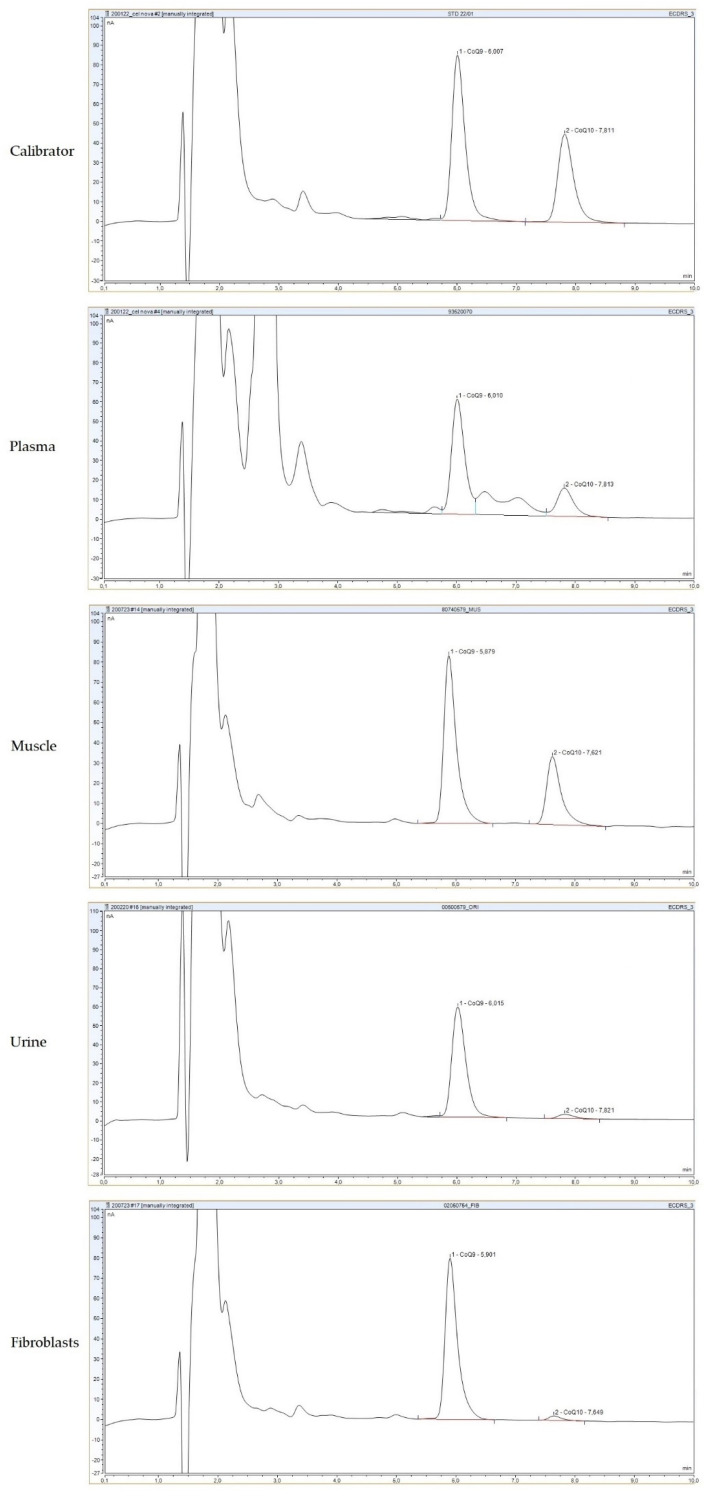
Typical chromatograms of CoQ determination by high-performance liquid chromatography with electrochemical detection (HPLC-ED) in different specimens: calibrator, plasma, muscle homogenate, urine and skin fibroblasts. CoQ_9_ was used as the internal standard.

**Figure 3 antioxidants-11-00528-f003:**
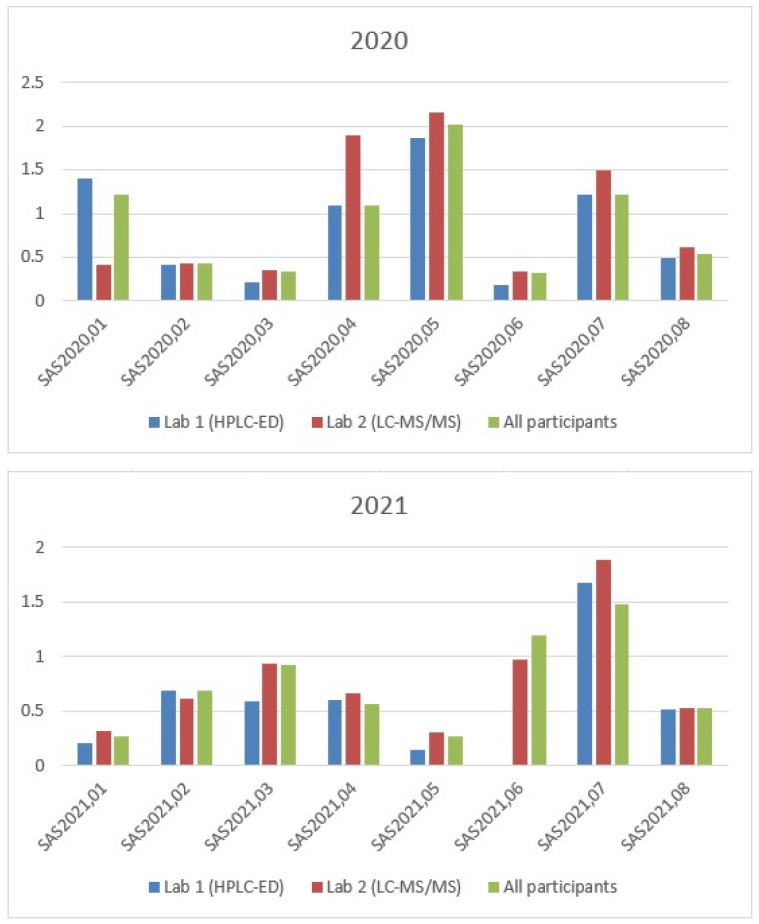
Visual comparison of the results of the EQC scheme for the plasma CoQ determination obtained with the HPLC-ED (Lab 1) and LC-MS/MS (Lab 2) methods, and from all participants during the 2020–2021 period. Mean values are stated for (Lab 1) and (Lab 2), while median values are reported for all participants. All data are expressed in µmol/L.

**Table 1 antioxidants-11-00528-t001:** Validation parameters of the new HPLC-ED method for CoQ analysis. Accuracy is expressed as % recovery, and its standard deviation is indicated in brackets. Abbreviations: AMR, analytical measurement range; LOQ, limit of quantification; LOD, limit of detection.

Validation Parameters	
Intra-assay precision (%)	
0.3 µmol/L	6.48
1 µmol/L	6.10
Inter-assay precision (%)	
0.74 µmol/L	8.85
Linearity (µmol/L)	0.06–7.07
*r* ^2^	0.999
AMR (µmol/L)	0.12–5.60
LOQ (µmol/L)	0.06
LOD (µmol/L)	0.02
Accuracy	100.6 (6.62)

**Table 2 antioxidants-11-00528-t002:** Global performance of all participants in the new external quality control (EQC) scheme for the plasma CoQ determination. For different metrological variables, mean values from all participants are provided, which were obtained from the annual reports during the 2020–2021 period. Abbreviations: CV, coefficient of variation.

	2020 (*n* = 11)	2021 (*n* = 13)
Mean values detected (µmol/L)	0.757	0.901
Precision (CV of the duplicates, %)	52.7	50.5
Linearity (*r*)	0.782	0.664
Recovery (% of added CoQ)	27	26
Interlaboratory CV (%)	141	82.2

**Table 3 antioxidants-11-00528-t003:** Results of the EQC scheme obtained with the HPLC-ED (Lab 1) and LC-MS/MS (Lab 2) methods, and from all participants. Mean values are stated for (Lab 1) and (Lab 2), while median values are reported for all participants. All data are expressed in µmol/L. Abbreviations: LC-MS/MS, liquid chromatography-tandem mass spectrometry; NR, not reported.

	Duplicate	Lab 1 (HPLC-ED)	Lab 2 (LC-MS/MS)	All Participants
SAS2020.01	1	1.4	0.41	1.2
SAS2020.02	2	0.41	0.43	0.43
SAS2020.03	3	0.22	0.36	0.34
SAS2020.04	4	1.1	1.9	1.1
SAS2020.05	4	1.9	2.2	2.0
SAS2020.06	3	0.18	0.33	0.32
SAS2020.07	1	1.2	1.5	1.2
SAS2020.08	2	0.49	0.61	0.54
SAS2021.01	1	0.21	0.32	0.27
SAS2021.02	2	0.69	0.62	0.69
SAS2021.03	3	0.59	0.94	0.92
SAS2021.04	4	0.60	0.67	0.57
SAS2021.05	1	0.14	0.30	0.27
SAS2021.06	3	NR	0.97	1.2
SAS2021.07	2	1.7	1.9	1.5
SAS2021.08	4	0.51	0.53	0.53

## Data Availability

The data are contained within the article and Appendix A.

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
