# Peer review of "Technical Aspects of Coenzyme Q10 Analysis: Validation of a New HPLC-ED Method"

_antioxidants, 2022, doi:10.3390/antiox11030528_

Round 1

Reviewer 1 Report

The manuscript by Artuch et al. provided a comprehensive technical result detailing the analysis of CoQ10 in a laboratory setting. The overall quality of the manuscript is high, with some minor issues that needs to be improved before publication. Details are below:

  1. To provide a better illustration of the compounds and general methodology, the authors should provide a chemical structure of CoQ10 as one of the figures and provide a flow-chart of the general analysis workflow with marks on the timing and technician level required. This will help readers to assess the application in the future.
  2. The authors should specify in the samples section what IRB protocol (or equivalent) was approved for use. The authors briefly mentioned the clinical sample issue in the discussion but did not provide details of the samples used in this study.
  3. In the results section, the authors stated that the method has been improved upon several optimization. However, the detailed performance was not provided. The authors should provide a figure similar to figure 1 as an illustration of the optimization step, which is significant according to the discussion part.
  4. The authors have explained why there is big variance in the results obtained from different labs. To help better understand the variance, could the authors provide comparison between old ED instrument with the new instrument proposed in the manuscript? A head-to-head comparison of the same sample would provide additional information in understanding how change of methods could impact the results.
  5. The authors may want to discuss the global gold-standard methods for measuring CoQ10 in various settings as part of introduction. A brief history of methodology advancement would also be helpful.
  6. The authors refer to HPLC as ‘high-pressure liquid chromatography’, however, the consensus refer to HPLC as ‘high performance liquid chromatography’. Unless the authors have special concerns or opinions, this should be changed to align with the general readers understanding.

Author Response

The manuscript by Artuch et al. provided a comprehensive technical result detailing the analysis of CoQ10 in a laboratory setting. The overall quality of the manuscript is high, with some minor issues that needs to be improved before publication. Details are below:

  1. To provide a better illustration of the compounds and general methodology, the authors should provide a chemical structure of CoQ10 as one of the figures and provide a flow-chart of the general analysis workflow with marks on the timing and technician level required. This will help readers to assess the application in the future.

Thank you for your suggestions, we agree with you and we have included two new supplementary figures: Figure S1, chemical structure of CoQ; Figure S2, flowchart of the general workflow for CoQ analysis in different biological samples, with marks on the timing required for a technician with HPLC skills.

  1. The authors should specify in the samples section what IRB protocol (or equivalent) was approved for use. The authors briefly mentioned the clinical sample issue in the discussion but did not provide details of the samples used in this study.

As we reported in the Institutional Review Board Statement (line 445), the study was approved by the Ethics Committee of the Hospital Sant Joan de Déu, and the reference of the protocol was PIC-97-16. Following the journal instructions for authors, we reported this information in this section and not in the samples section. In order to provide more details of the samples used in the study, we have included a new sentence in section 2.1. (Samples) of the manuscript: ‘Samples from patients with suspected and confirmed plasma CoQ deficiency and controls were used for this study.’

  1. In the results section, the authors stated that the method has been improved upon several optimization. However, the detailed performance was not provided. The authors should provide a figure similar to figure 1 as an illustration of the optimization step, which is significant according to the discussion part.

Thanks for your comment. In order to better illustrate the optimization of the method, we have included a new supplementary figure (Figure S3). Figure S3a depicts the analysis of a calibrator solution after we changed the composition of the mobile phase and before complete optimization of the method. In this case, a gradual loss in cell performance was observed, as indicated by the decreasing peak heights of CoQ. Figure S3b shows the analysis of a pool of plasma samples with the further optimized method, which included an electrochemical treatment of cells after each run; in this case, cell performance was stable in time.

  1. The authors have explained why there is big variance in the results obtained from different labs. To help better understand the variance, could the authors provide comparison between old ED instrument with the new instrument proposed in the manuscript? A head-to-head comparison of the same sample would provide additional information in understanding how change of methods could impact the results.

As described in section 3.1. (line 220), we did compare our old HPLC-ED method with the new one using 54 samples and Passing-Bablok regression analysis. We have included a new supplementary figure (Figure S4) with these results, as also suggested by reviewer 2.

  1. The authors may want to discuss the global gold-standard methods for measuring CoQ10 in various settings as part of introduction. A brief history of methodology advancement would also be helpful.

Classically, HPLC with both UV and ED detection have been the gold standard. ED has advantages, especially for sensitivity and the simultaneous detection of oxidized and reduced species. Undoubtedly, in a near future, mass spectrometry techniques will be available in most clinical laboratories and they will become the reference method for this purpose. We have modified the introduction in order to include these aspects suggested by you and reviewer 2.

  1. The authors refer to HPLC as ‘high-pressure liquid chromatography’, however, the consensus refer to HPLC as ‘high performance liquid chromatography’. Unless the authors have special concerns or opinions, this should be changed to align with the general readers understanding.

We agree with you, and we have modified the original abbreviation according to your suggestion.  

Reviewer 2 Report

This is an interesting work that presents a development of a new method for CoQ10 analysis. The test has a high significance to monitor certain pathological conditions. I am disappointed by a lack of explanation for the analytical data. There are certain points that must be added and discussed in terms of method validation and performance. There is a clear consensus in laboratory medicine how new method characteristics should be presented for scientific community and I will encourage authors to follow this consensus. See more specific points below:

In the introduction paragraph, please add more details why it is significant to measure CoQ10 clinically. Perhaps moving lines 246-255 to the introduction and add references when describing pathological condition associated with the CoQ10.

Line 51: Add more references and details on current CoQ10 method. Foe example, I found more publications on LC-MS/MS or GCMS methods. Also, please add typical wavelength for UV detection.

Line 92: Is this one calibrator (1.16 μmol/L ) or calibrators?

Line 199: “Metrological”  variables. Please find a more appropriate description of measured variables

Section 3.2 method validation- it will look more clear if authors would report method performance characteristics in a table format.

What is the analytical measurement range (AMR) for the developed method and what are medical decision limits (If exist)?

What was the solvent of internal standard (CoQ9)?

Line 115: It is not clear to me how 20 plasma samples were prepared? Did authors spiked 20 different plasma samples (or samples are from the same pool) with two levels of CoQ10 (0.3 μmol/L and 1 μmol/L). This needs to be clarified.

Line 185: please include Passing-Bablok regression analysis as one of the figures

Table 1: please add standard deviation when you report accuracy. It is not clear whether data presented in Table 1 belongs to the new method? If the method performance characteristics represent a new method, authors need to discuss why there is a such low recovery and linearity.

Line 200: why two specific concentrations 0.3 and 1 umol/ml were chosen for validation. Are these concentrations serve as low and high controls?

Line 202: Why 0.74 umol/L was chosen to study inter-assay precision?

Please report limit of detection (LOD) of your method

How LOQ (equation) was calculated?

Did you study matrix effect? What was the recovery in different matrices? Calibrators were prepared in ethanol and not biological matrix. While often we don’t have a commercial source appropriate matrix,  this study limitation needs to be discussed.

Accuracy of all methods summarized  (Table 1) should be reported in a different way.

Author Response

This is an interesting work that presents a development of a new method for CoQ10 analysis. The test has a high significance to monitor certain pathological conditions. I am disappointed by a lack of explanation for the analytical data. There are certain points that must be added and discussed in terms of method validation and performance. There is a clear consensus in laboratory medicine how new method characteristics should be presented for scientific community and I will encourage authors to follow this consensus. See more specific points below:

We agree with you and, in fact, we have validated different procedures following the standard parameters required by scientific societies. In this case, since our previous HPLC-ED method was already accredited by UNE-EN ISO 15189 norm for medical laboratories, we believed that the validation of the new one could relay on the basic parameters that we have reported plus the external quality control assessment. However, we agree with you and we have added new parameters following your suggestions. Thank you.

  1. In the introduction paragraph, please add more details why it is significant to measure CoQ10 clinically. Perhaps moving lines 246-255 to the introduction and add references when describing pathological condition associated with the CoQ10.

Ok, we have moved lines 246-255 to the introduction and we have added new references.

  1. Line 51: Add more references and details on current CoQ10 method. For example, I found more publications on LC-MS/MS or GCMS methods. Also, please add typical wavelength for UV detection.

As also suggested by reviewer 1, we have added more details regarding methods for CoQ analysis, including more references and a brief history of methodology advancement.

  1. Line 92: Is this one calibrator (1.16 μmol/L) or calibrators?

Sorry, we did probably not properly explain this issue: we used one calibrator and one control. We have modified the manuscript in order to clarify this.

  1. Line 199: “Metrological”  variables. Please find a more appropriate description of measured

Ok, we have replaced ‘metrological variables’ by ‘analytical parameters’.

  1. Section 3.2 method validation- it will look more clear if authors would report method performance characteristics in a table format.

Ok, we have modified section 3.2. in order to report these results in a table format (new Table 1).

  1. What is the analytical measurement range (AMR) for the developed method and what are medical decision limits (If exist)?

Beyond linearity we have calculated AMR as the range of concentrations that the method can directly measure without any dilution, concentration, or other pre-treatment (see new Table 1). AMR was assessed by analyzing ten times five plasma samples covering very low plasma CoQ values (0.12 µmol/L), moderate deficiency (0.3 µmol/L), normal values (0.74 µmol/L and 1 µmol/L), and very high values due to CoQ supplementation (5.6 µmol/L). All of them showed a CV lower than 10%.

Regarding medical decision limits, there is not a general consensus. In any case, we can consider the lowest limit of the reference interval (around 0.4 µmol/L) the limit to detect plasma CoQ deficient status. This value is inside the AMR. Regarding the highest CoQ values, there is not a clear medical decision limit since these values are only observed when patients are under CoQ supplementation. Once again, there is not a consensus about which plasma CoQ values are advisable for treatment monitoring. We have added a new statement considering these issues in the discussion section.

  1. What was the solvent of internal standard (CoQ9)?

As reported in line 117, the solvent was ethanol.

  1. Line 115: It is not clear to me how 20 plasma samples were prepared? Did authors spiked 20 different plasma samples (or samples are from the same pool) with two levels of CoQ10 (0.3 μmol/L and 1 μmol/L). This needs to be clarified.

We agree with you that this point could be better explained, and we have modified the manuscript in order to clarify this. For the intra-assay precision study, we used two different pools, each one with different CoQ concentrations (0.3 and 1 µmol/L) close to the low and high limits of our reference interval. Samples were not spiked, we used plasma samples with known CoQ values to prepare the pools.   

  1. Line 185: please include Passing-Bablok regression analysis as one of the figures

We have included a new supplementary figure (Figure S4) with the results of Passing-Bablok regression.

  1. Table 1: please add standard deviation when you report accuracy. It is not clear whether data presented in Table 1 belongs to the new method? If the method performance characteristics represent a new method, authors need to discuss why there is a such low recovery and linearity.

Sorry, but data reported in Table 1 (which now is Table 2 in the revised manuscript) does not belong to the new method but to the annual performance for all participants of the EQC scheme. Our validation parameters are now stated in the new Table 1 of the revised manuscript, as you suggested, and we have included standard deviation for accuracy.

  1. Line 200: why two specific concentrations 0.3 and 1 umol/ml were chosen for validation. Are these concentrations serve as low and high controls?

These two concentrations were chosen as they are close to low and high plasma CoQ values, in accordance with our reference intervals.

  1. Line 202: Why 0.74 umol/L was chosen to study inter-assay precision?

Because this concentration is the commercially available and, since either low and high values were used for intra-assay precision study, we thought convenient to use an intermediate value for assessment of inter-assay precision, thus covering all the range of CoQ values (low, normal, high).

  1. Please report limit of detection (LOD) of your method.

We have now reported LOD of the method (see new Table 1), and also how it was calculated by adding a new sentence in section 2.3.

  1. How LOQ (equation) was calculated?

LOQ was calculated as (10*SD of the response)/slope of the calibration curve. We have added a new sentence in section 2.3. regarding this issue.

  1. Did you study matrix effect? What was the recovery in different matrices? Calibrators were prepared in ethanol and not biological matrix. While often we don’t have a commercial source appropriate matrix, this study limitation needs to be discussed.

No, we did not study matrix effect. There are not commercially available calibrators for CoQ in other matrices rather than ethanol, since CoQ solubility is a critical issue. However, control material is already prepared in a matrix similar to real samples, as it is based on human plasma. On the other hand, we agree with you that this is a potential limitation and we have added a new paragraph in the discussion section about the limitations of our study. In fact, we have modified the title of the manuscript, as also suggested by the reviewer 3.

  1. Accuracy of all methods summarized (Table 1) should be reported in a different way.

We do not have the data of all participants in the EQC scheme since this information is confidential. However, as suggested by you, we have added the comparison of our accuracy with that of the EQC scheme and we have expressed it in a different way (as % of deviation) (see new Supplementary Table 1). Moreover, we have renamed this variable as mean values reported by all participants in the scheme in Table 2, as we agree that it can lead to certain confusion.

Reviewer 3 Report

In the manuscript “Technical aspects of coenzyme Q10 analysis in different biological samples” by Abraham J. Paredes-Fuentes et al. the authors describe a new method based on HPLC with electrochemical detection (ED) for CoQ determination. Even if the electrochemical detectors is one of the most used detectors for the  HPLC analysis of CoQ levels in biological samples, the standardization of a new procedure is justified because the production of the electrochemical cells used with the Coulochem series detectors was discontinued. In this contest, the main aims of the authors were to standardize the procedure with an instrument equipped with a new electrochemical cells and to participate to  a new external quality control (EQC) scheme for the plasma CoQ determination comparing their results with those obtained with an LC- MS/MS method.

Plasma CoQ determination, being minimally invasive, is particularly useful in monitoring CoQ levels in patients affected by CoQ deficiency. However, the degree and mode of distribution of plasma CoQ to tissues is not completely known, and is still a matter of debate. The clinical assessment of CoQ deficiency based on CoQ level determination is complicated by the large reference range for plasma CoQ10 values that varies from approximately 0.26 to 1.7 μM and this wide range of values suggests that plasma CoQ content may be affected by multiple factors. Moreover, there are conflicting data in the literature regarding ubiquinone levels measured in tissue biopsies or plasma samples, and that the quantification and normalization of such data are subject to experimental limitations. For this reason, a validation procedure for CoQ determination and a statistical analysis of the data obtained in different laboratory is important.

The validation procedure presented here looks accurate and the results were good an in line with the HPLC-MS/MS procedure. As assessed by the authors, there are some issues that has to be improved such as the analyte recovery and the inter-laboratory variations.

The only concern that can be addressed regards the title: the validation procedure is limited to plasma samples and not to “different biological samples” so I just suggest to modify the title.

Author Response

In the manuscript “Technical aspects of coenzyme Q10 analysis in different biological samples” by Abraham J. Paredes-Fuentes et al. the authors describe a new method based on HPLC with electrochemical detection (ED) for CoQ determination. Even if the electrochemical detectors is one of the most used detectors for the  HPLC analysis of CoQ levels in biological samples, the standardization of a new procedure is justified because the production of the electrochemical cells used with the Coulochem series detectors was discontinued. In this contest, the main aims of the authors were to standardize the procedure with an instrument equipped with a new electrochemical cells and to participate to a new external quality control (EQC) scheme for the plasma CoQ determination comparing their results with those obtained with an LC- MS/MS method.

Plasma CoQ determination, being minimally invasive, is particularly useful in monitoring CoQ levels in patients affected by CoQ deficiency. However, the degree and mode of distribution of plasma CoQ to tissues is not completely known, and is still a matter of debate. The clinical assessment of CoQ deficiency based on CoQ level determination is complicated by the large reference range for plasma CoQ10 values that varies from approximately 0.26 to 1.7 μM and this wide range of values suggests that plasma CoQ content may be affected by multiple factors. Moreover, there are conflicting data in the literature regarding ubiquinone levels measured in tissue biopsies or plasma samples, and that the quantification and normalization of such data are subject to experimental limitations. For this reason, a validation procedure for CoQ determination and a statistical analysis of the data obtained in different laboratory is important.

The validation procedure presented here looks accurate and the results were good an in line with the HPLC-MS/MS procedure. As assessed by the authors, there are some issues that has to be improved such as the analyte recovery and the inter-laboratory variations.

Thank you very much for your positive comments.

The only concern that can be addressed regards the title: the validation procedure is limited to plasma samples and not to “different biological samples” so I just suggest to modify the title.

We agree with you, in fact reviewer 2 raised a similar concern and in this sense we have added a new paragraph in the discussion section regarding the limitations of our study, focused mainly in the fact that the validation was done for plasma samples, but not for the other biological specimens. Thus, we have also modified the title of the manuscript for ‘Technical aspects of CoQ analysis: validation of a new HPLC-ED method’.

Round 2

Reviewer 2 Report

The manuscript is significantly improved now. There is one minor edit that I request. In Table 1: Please indicate units (I think umol/ml?) in the linearity range (0.06-7.07) and round r2 to the three significant figures.